# OpenReview forum: "Monitoring LLM Agents for Sequentially Contextual Harm"
_ICLR.cc/2025/Workshop/BuildingTrust — BuildingTrust_

### Official Review · Reviewer_hGNQ · 2025-03-01
**Well-Structured Study on Sequentially Contextual Harm in LLM Agents**

**Rating:** 8
**Confidence:** 3

**Review:**

Summary:
The authors investigate a vulnerability in LLM safety systems: the ability of users to bypass harm detection by breaking malicious requests into individual benign subtasks. Their paper demonstrates that refusal rates drop dramatically when high-level malicious prompts are broken down into individual benign prompts. However, LLMs were still able to identify user intent reasonably well. The authors identify four key takeaways:  LLMs struggle to connect benign subtasks to harmful overall goals, unrelated innocent tasks can be strategically inserted to further mask malicious intent, conventional prompt engineering proves inadequate in addressing this vulnerability, and the capability of models is not correlated with their performance on these tasks. These support the authors’ call for more specialized approaches to develop more robust agent monitoring systems.

Originality and Significance:
While prior research has explored breaking malicious prompts into steps, this paper is the first to explore this phenomenon in agentic LLM settings, along with a thorough empirical analysis, making a valuable contribution to the field.

Pros:

- Evaluated on a wide range of different LLMs
- The paper provides a novel formulation of using task decomposition to study sequentially contextual harm in LLM agents
- Strong experimental results with detailed metrics to support the sequentially contextual harm formulation and demonstrated inadequacy of current methods for monitor
- Paper is well organized and easy to follow

Cons:

- Limited dataset scope → only 50 agent tasks covering 8 categories
- Methodology for benign subtask creation: How is “benign” operationalized and validated?

---

### Official Review · Reviewer_GcxG · 2025-03-01
**Interesting insights about sequentially Contextual Harm Monitoring. Greater Clarity on Motivation and System Setting would be a plus.**

**Rating:** 6
**Confidence:** 4

**Review:**

Pros:
1. The paper presents interesting findings on how malicious tasks can be decomposed into seemingly benign subtasks that, when executed sequentially, ultimately yield harmful outcomes. The comparison among single-turn tasks, multi-turn subtasks, and single-turn subtasks effectively highlights real-world scenarios of how an agent might receive user requests.

2. Substantial effort has gone into curating and augmenting a decomposed task dataset, building on AgentHarm benchmarks. This work adds valuable scope and diversity to the existing datasets.

3. The authors carefully design several LLM-based monitoring setups, offering insights on the impact of including safety guidelines, the effectiveness of ensemble approaches, and other factors relevant to monitor performance.

Cons:
1. The paper does not clearly introduce or specify the agent system used. Different agent frameworks may have distinct designs and respond differently to malicious tasks. Clarifying the system and potentially evaluating multiple agent systems would strengthen the paper.

2. Using LLMs as a monitoring model has already shown promise in pure language-model settings. While applying it to an agentic environment is useful, it could be seen as a direct extension rather than a novel conceptual contribution.

3. Although the authors invested considerable effort in creating high-quality in-context samples for GPT-4o to generate additional subtask decompositions, the process remains partly human-driven. The paper would benefit from demonstrating that these curated and filtered subtasks accurately reflect how an agent might naturally break down complex requests into multiple steps. Otherwise, the settings are similar as testing on standalone LLMs understanding contextual semantic harmfulness.


Overall: I recommend accepting the paper. The dataset and experiments are carefully designed, and the research topic is timely, given the rapid growth of agent-based systems.

---

### Official Review · Reviewer_MWzx · 2025-03-02
**The paper addresses AI agents' vulnerability to subtle sequential harm, but could benefit from a larger dataset and exploring additional monitoring techniques for more comprehensive countermeasures.**

**Rating:** 7
**Confidence:** 3

**Review:**

The paper highlights the crucial issue of AI agents being vulnerable to seemingly benign subtasks, and the authors used rigorous experiments on different LLMs to evaluate and demonstrate that sequential contextual harm is a safety challenge in LLM agents.

I think increasing the dataset would have further strengthened the paper. It only examines a few basic techniques to improve monitoring, and investigating other methods could have provided more comprehensive countermeasures to LLM-agent vulnerabilities.

The paper is well-written and a good fit for the workshop.

---

### Decision · Program_Chairs · 2025-03-04

Accept